# Beyond What Your Retina Can See: Similarities of Retinoblastoma Function between Plants and Animals, from Developmental Processes to Epigenetic Regulation

**DOI:** 10.3390/ijms21144925

**Published:** 2020-07-12

**Authors:** Estephania Zluhan-Martínez, Vadim Pérez-Koldenkova, Martha Verónica Ponce-Castañeda, María de la Paz Sánchez, Berenice García-Ponce, Sergio Miguel-Hernández, Elena R. Álvarez-Buylla, Adriana Garay-Arroyo

**Affiliations:** 1Laboratorio de Genética Molecular, Epigenética, Desarrollo y Evolución de Plantas, Instituto de Ecología, Universidad Nacional Autónoma de Mexico, 3er Circuito Ext. Junto a J. Botánico, Ciudad Universitaria, UNAM 04510, Mexico; ezluhanm@gmail.com (E.Z.-M.); mpsanchez@iecologia.unam.mx (M.d.l.P.S.); bgarcia@ecologia.unam.mx (B.G.-P.); 2Posgrado en Ciencias Biomédicas, Universidad Nacional Autónoma de México, Av. Universidad 3000, Coyoacán 04510, Mexico; 3Laboratorio Nacional de Microscopía Avanzada, Centro Médico Nacional Siglo XXI, Instituto Mexicano del Seguro Social, Av. Cuauhtémoc, 330. Col. Doctores, Alc. Cuauhtémoc 06720, Mexico; vadim.perez@imss.gob.mx; 4Unidad de Investigación Médica en Enfermedades Infecciosas, Centro Médico Nacional SXXI, Instituto Mexicano del Seguro Social, Mexico City 06720, Mexico; vponce@ifc.unam.mx; 5Laboratorio de Citopatología Ambiental, Departamento de Morfología, Escuela Nacional de Ciencias Biológicas, Instituto Politécnico Nacional, Campus Zacatenco, Calle Wilfrido Massieu Esquina Cda, Manuel Stampa 07738, Mexico; sergio.mi.encb@gmail.com

**Keywords:** retinoblastoma protein, cell proliferation, cell differentiation, plants, animals, cell cycle, stem cells, epigenetics, DNA damage, morphogenetic regulatory networks

## Abstract

The Retinoblastoma protein (pRb) is a key cell cycle regulator conserved in a wide variety of organisms. Experimental analysis of pRb’s functions in animals and plants has revealed that this protein participates in cell proliferation and differentiation processes. In addition, pRb in animals and its orthologs in plants (RBR), are part of highly conserved protein complexes which suggest the possibility that analogies exist not only between functions carried out by pRb orthologs themselves, but also in the structure and roles of the protein networks where these proteins are involved. Here, we present examples of pRb/RBR participation in cell cycle control, cell differentiation, and in the regulation of epigenetic changes and chromatin remodeling machinery, highlighting the similarities that exist between the composition of such networks in plants and animals.

## 1. Introduction

Eukaryotic organisms evolved from a common unicellular ancestor, from which they diverged 1500 million years ago [1,2,3]. Testimonies of this ancient kinship are some processes and genetic components that have been preserved throughout evolution, along with many others which subsequently emerged exclusively in plants or animals [4,5,6]. Interestingly, it has been shown that plants and animals evolved multicellularity independently, but unlike animals, plants have an extensive post-embryonic growth and development, that is highly influenced by the environment [7,8]. However, in both multicellular organisms, morphogenesis depends on a delicate balance between cell division and differentiation rates. In animals, when this balance is lost, it can cause tumors and cancer, not seen during most of plant development. Nonetheless, *Arabidopsis thaliana* (hereafter Arabidopsis), the most studied model plant, has been shown to be an important model system to understand basic regulatory mechanisms involved in human diseases [9,10,11]. For example, Arabidopsis has homologous genes for 70% of those involved in human cancer. Interestingly, a higher percentage than that found in the genome of *Drosophila melanogaster* or of *Saccharomyces cerevisiae* [9,10]. Hence, Arabidopsis has already been used as a screening tool to evaluate the action and efficacy of some drugs to treat human cancer and other diseases [11,12,13].

Retinoblastoma (*RB1*) is a highly conserved gene, that encodes the so-called tumor suppressor protein, that regulates different developmental processes of phylogenetically distant organisms. The name of this gene comes from the retinoblastoma illness, a rare intraocular malignant tumor that has its onset during early childhood. As determined by karyotyping, this disease is associated with losses in chromosome 13, which contains the *RB1* gene [14,15,16]. *RB1* was identified by positional cloning and after subsequent molecular analysis, it became known as the first tumor suppressor gene, giving robust evidence for the genetic predisposition of cancer development in some cases [17,18]. After its discovery, alterations in this gene were described in other malignant tumors such as osteosarcomas, cervical cancer, prostate carcinoma, small cell lung cancer, and some forms of leukemia [19,20]. *RB1* is an essential gene whose best studied function is the regulation of the cell cycle transition from G1 to S phase through formation of a protein complex with transcription factors of the E2F-family; that are regulated by the Retinoblastoma protein (pRb) multiple phosphorylation states. In many cancer types, an altered regulation of pRb, like permanent hyperphosphorylation that promotes pRb dissociation from the complex with E2F, leads to an unregulated cell proliferation [21,22]. Moreover, altered regulation of the pRb pathway is considered one of the most common traits in different types of cancer [23,24], and several studies have proposed targeting pRb regulation pathway as alternative treatments [25,26,27]. In fact, cyclin dependent kinases (CDKs), the kinases that phosphorylate pRb are commonly deregulated in many malignant tumors. From the therapeutic standpoint, pRb cannot be a target of drugs, however, CDKs are therapeutic targets, and several generations of non-specific cell cycle CDKs inhibitors have been under clinical evaluation as cancer treatments with mixed results. More recently specific cell cycle CDK4/6 and transcriptional CDKs inhibitors may become alternative therapeutic strategies under current clinical evaluation [28,29,30]. In summary, a more thorough understanding of pRb’s developmental functions could help find new efficient treatments for different cancer types.

In this review, we will focus on how the protein encoded by the *RB1* gene, and its plant ortholog *RETINOBLASTOMA-RELATED* (*RBR*), participates in important developmental processes such as cell cycle control, cell differentiation, as well as in the homeostasis of stem cell/pluripotency, that are cellular processes shared by plants and animals. First, we analyze the structure of the RB/RBR proteins, secondly, we describe their interaction with proteins that participate in different developmental processes and regulatory networks. Finally, we analyze their participation in the epigenetic and chromatin structure regulation, which is important for different developmental decisions. Interestingly, many of the protein interactions involving RETINOBLASTOMA, as well as the overall structures of the regulatory networks in which this protein participates are similar between plants and animals. This suggests that once such regulatory networks were assembled during evolution, the key role of this protein as an integrator of internal developmental cues remained functionally constrained among eukaryotic organisms’ evolution.

## 2. Structure of the Retinoblastoma Protein

In humans, Retinoblastoma susceptibility gene is a member of a small gene family that includes *RB1* (p105/pRb), *RBl1* (p107/pRBL1), and *RBl2* (p130/pRBL2), whose protein structure are very similar, and that share some overlapping functions [31,32,33]. From the three family members, *RB1* has been the most studied gene since it participates in tumor onset and progression, while *RBl1* and *RBl2* rarely display mutations in human retinal cancer [34,35]. 

The human Retinoblastoma protein (pRb) consists of 928 amino acids and includes three distinctive domains: the N-terminal structural domain (RbN), the so-called “pocket” (RbP) domain, the C-terminal domain (RbC), and the non-structured regions between them (Figure 1A). The pocket domain includes two highly conserved subdomains (A and B) called cyclin folds, which are formed by two structural nuclei, each conformed by three helix bundles with two additional helices packing at the sides in each one. These subdomains are required to mediate interactions with other proteins like several oncoproteins and transcription factors (TFs) [34,36,37,38]. According to current interaction databases 322 proteins interact with human pRb, the E2F TFs being the best characterized ones (Figure 1A) [39,40]. The interaction of pRb with many other proteins depends on the pRb structure and its post-translational modifications, which determine this protein’s function in different developmental processes [40,41]. Many of the pRb-interacting proteins contain the motif ‘LxCxE’ (Leu-X-Cys-X-Glu where X stands for any amino acid), essential to bind with the Pocket B subdomain of pRb (Figure 1A). Examples of such proteins are D-type cyclins that are cell cycle regulators, the histone deacetylases 1 and 2 (HDAC1/2); several viral proteins like SV40 large T antigen (SV40 T-ag) and two viral proteins that stimulate the cell cycle progression in infected cells through pRb inactivation: Human Papillomavirus E7 (HPV E7) and Adenovirus early region 1A (Ad5 E1A) [38,42,43,44]. The RbN domain is also composed of two cyclin folds very similar to those found in the “pocket” domain. The RbN domain can physically interact with the RbP one and deletion of this domain abrogates the regulation of the pRb/E2F complex [45]. Finally, the RbC region, that includes approximately 150 residues, is intrinsically disordered and has been shown to be required for the interaction between pRb and the E2F/DP complex [36,46]. The three pRb domains are connected by sequences that confer flexibility to the protein and contain sites that can be subjected to post-translational modifications, that play important roles in the regulation of pRb activity [47]. 

The post-translational modifications can assist or avoid occurrence of other modifications that, together, can modulate pRb protein interactions and function and create a diversity of biological activities, making this protein a key node in several regulatory networks. In total, human pRb has 14 phosphorylation sites [37,40], two acetylation sites [43,48], six methylation sites [49,50,51], and it can also be modified through ubiquitination or sumoylation [52,53,54] (Figure 1B). Studies on the structure of pRb have revealed that phosphorylation changes pRb structure and promotes new interactions with other proteins. For example, phosphorylation of residues S_608_/S_612_, localized between subdomains A and B of the Pocket domain, induces a new conformation of the pocket domain and its interdomain, that partially prevents E2F binding to pRb. Phosphorylation of residue T_373_ promotes the binding of the RbN domain to the Pocket domain. The result is a globular structure of pRb that can no longer interact with E2F. In addition, phosphorylation of the S_788_/S_795_ and T_821_/T_826_ residues, affects the interactions of the RbC domain with the E2F/DP complex [45,46,47,55,56]. Furthermore, acetylation and methylation sites are close to the carboxyl-terminal of the protein. Introduction of these post-translational modifications depend upon different signals such as pRb localization, DNA damage, and cell differentiation [48,51,57]. 

The *RB* ortholog in plants was identified approximately a decade after the animal gene was discovered: *RB1* gene was first described in humans between 1986 and 1989 [17,18,58] and the first plant orthologous *RB1* cDNA (*RBR*), was first identified and cloned from maize in 1994 (*ZmRBR*) [59,60,61], then in tobacco (*NtRBR*) [62] and then in Arabidopsis (*AtRBR*) [63]. Afterwards, there have been numerous reports characterizing *RBR* orthologs in different plant species. Intriguingly, monocotyledons seem to have various *RBR* paralogs while dicotyledons have only one [64]. As a dicot plant, Arabidopsis carries a single copy of the *RBR* gene displaying ≈35% sequence similarity within the pocket domain with respect to the human pRb. Interestingly, and even though the function of the N-terminal domain has not yet been characterized, this region is highly conserved between pRb from mammals and the RBR from *Zea mays* and Arabidopsis [65,66,67], suggesting that this region could also be involved in a conserved function in plants and animals. The Arabidopsis RBR (AtRBR) protein contains 1013 amino acids with the same modular structure of human pRb with putative phosphorylation residues. The role of four of them, located in the protein’s inter-domains, have been experimentally tested (Figure 1B) [68,69,70]. Interestingly, we observe that the S_685_ phosphorylation site in AtRBR is conserved in the same interdomain, between RbP A and RbP B subdomains, found in pRb (S_608_/S_612_), suggesting a conserved function in mediating the interaction with E2F TFs in animals and plants (Figure 1B). Additionally, human antibodies for human phospho-pRb protein in S_807_/_811_ can bind to RBR of *Medicago sativa* and Arabidopsis [71,72]. Even though the specific regulatory functions of these sites are still unknown, it has been shown in plants that cyclin-dependent kinases also phosphorylate RBR to regulate cell cycle through its inactivation and release of E2F, similar to what it has been described in humans [73,74]. 

Like in humans, in plants there are viral proteins (e.g., RepA from geminivirus) that also have the ability to interact with the pocket domain of RBR [61,75,76]. This suggests that the protein–protein interaction between pRb/RBR with specific viral proteins is a viral mechanism that controls cell cycle progression both in plants and animals. Such viral–eukaryotic cell interaction could have been established before plant and animal divergence, or it has evolved independently.

pRb is also found in unicellular organisms such as the algae *Chlamydomonas reinhardtii*, the choanoflagellate *Monosiga brevicollis*, and the amoeba *Dictyostelium discoideum*, suggesting that it was already present in the common eukaryotic ancestor before the separation of animal and plant lineages, before they diverged. In addition, at least one copy of *RB* has been identified in species from each of the eukaryotic supergroups [68,77]. 

## 3. Cell Cycle Control through Retinoblastoma

The cell cycle is a well-studied process essential for the growth and reproduction of all eukaryotic organisms. It assures the faithful duplication of the genetic material and its distribution between two daughter cells [78,79]. The cell cycle has two major phases: Interphase and Mitosis phase. Additionally, in the interphase other three stages can be distinguished: G1, S, and G2. Many of the components that regulate these phases and the transition between them, are conserved among different organisms [80,81]. The availability of growth factors, nutrients, and intrinsic developmental signals determine whether a cell remains in a quiescent state, when the cell does not divide (G0), or transits from phase G1 to S, during which the genetic material is duplicated to later divide. In humans, this transition is under the control of the pRb/E2F pathway, that regulates the transcription of genes encoding proteins involved in DNA synthesis [22]. 

The G1/S transition is one of the main regulation checkpoints of the cell cycle, being the “commitment point”, also known as the “restriction point” in animals, the one that determines the cell’s commitment to engage proliferation in a way independent from environmental signals [82,83]. Therefore, at this point the cell integrates environmental and intrinsic signals to prepare its nucleus to start cell division. A deregulation of this transition in humans can lead to the generation of tumors and cancer [22,84,85]. pRb hypophosphorylated acts as a negative regulator of cell cycle progression through its interaction with the E2F proteins. The heterodimer keeps the chromatin in a closed conformation in the regulatory regions of E2F-regulated genes [22,86]. The E2F family includes the transcription factors E2F1, E2F2, and E2F3a, which typically promote transcription and E2F3b, E2F4-E2F8 that are associated with transcriptional repression. However, it has been reported that, depending on the developmental stage, some E2Fs, like E2F1-4, can function both as activators or as repressors of transcription [87,88,89]. E2F1-6 members can heterodimerize with the Dimerization Partner (DP) proteins, although this interaction is not always required for transcriptional activation [90]. E2F7 and E2F8 are independent of DP and these TFs are also different to other E2Fs because they possess two DNA-binding domains [87,91,92]. The hypophosphorylated pRb form binds and inhibits activating E2F1-3, whereas E2F4 and E2F5 bind to pRBL1 (p107) and pRBL2 (p130) at the promoters of target genes, to repress transcription [22,93]. In addition, E2F1-3 carry a nuclear localization signal, whereas E2F4 and E2F5 lack it, and apparently rely on pRBL1 and pRBL2 for their nuclear translocation. On the other hand, E2F6-8 lacks the sequences required to bind with pRbs [94,95,96]. At the same time, the expression of different E2Fs is itself subjected to spatiotemporal regulation during cell cycle progression and different stages of development [97]. Moreover, when the pRb/E2F interaction is disrupted by loss or reduction of pRb, a high rate of cell proliferation is observed, and this generally triggers cancer [98].

pRb phosphorylation by cyclin/CDKs (cyclin-dependent kinases) complexes changes the pRb protein structure and its interactions with other proteins, inducing the release of E2F. For instance, this occurs in human cells when cyclin type D or E (CYCD/E) associate with cyclin kinases 4 or 2 (CDK4/2), respectively (Figure 2A) [22,99,100]. When pRb phosphorylation is altered, for example by overexpression of CYCDs, or by a miss regulation of CDKs, or by a disruption of the LxCxE-binding function the cell cycle is altered [101,102,103].

In humans, pRBL1 and pRBL2 participate in the repression of genes when cells are in the G0 quiescent state, through a complex called DREAM (DP-Rb-E2F-MuvB), that coordinates the repression of genes during quiescence and also the periodic gene expression during the G1/S and G2/M transitions [104,105]. DREAM is a multimeric protein complex that in humans is composed of DP (DP1-DP3), pRBL1 or pRBL2, E2F proteins (E2F4 and E2F5), and the subcomplex MuvB (Multi vulval class B). The MuvB subcomplex acts as a repressor when it is part of the DREAM complex, and is composed of LIN (LIN9, LIN37, LIN52, LIN54) and RBBP4 proteins. When pRb is phosphorylated, the DREAM complex is disassembled and the MuvB subcomplex can associate with TFs such as B-Myb (Myb type) and Fox-M1 (Forkhead box protein M 1), to promote the regulation of gene expression and the transition from quiescence to proliferation (Figure 2A) [104,106,107]. This complex is also conserved in phylogenetically distant organisms, but some of its components can vary among species, as in Arabidopsis compared to mammals (see below).

At about the same time that the pRb ortholog in plants was discovered, homologs of other components of the animal’s cell cycle regulatory machinery were identified and characterized in corn, as well as in Arabidopsis and *Medicago sativa* (alfalfa) [61,108,109,110,111]. In these plants, it was shown that the phosphorylation of RBR by the CDKA/CYCD protein complex regulates cell cycle progression, as it occurs in humans (Figure 2B) [73,74].

In Arabidopsis, the null mutant plants of *AtRBR* are gametophytic lethal because of supernumerary nuclei alterations in the divisions at late stages of female gametogenesis; whereas the male gametophyte (pollen) contains multiple sperm cells [112,113]. Therefore, in order to study the function of *RBR* function of Arabidopsis and maize, either its transcript accumulation has been reduced by RNAi or conditional repression has been employed. In addition, protein competence for RBR binding, has also been used to address the role of RBR in cell proliferation [114,115,116,117]. 

In monocots, like rice, wheat, barley, and sugarcane, at least two different RBR types are present, RBR1 and RBR3 [64], although maize carries four ZmRBR genes: ZmRBR1, ZmRBR3, and two paralogs, ZmRBR2 and ZmRBR4 [118,119]. ZmRBR1 is constitutively expressed, and its protein interacts with E2F and with a HDAC (ZmRpd3I) that lacks the characteristic LxCxE motif. Besides, the two paralogs, ZmRBR1/2, negatively regulate cell cycle as it occurs in humans and Arabidopsis (Figure 2C) [119,120,121,122]. In addition, ZmRBR3/4 transcripts accumulate in mitotic tissues from the endosperm. Recently, high levels of ZmRBR3/4 were found in tumor-like formations (induced by the fungus *Ustilago maydis*) from maize leaves [118,123]. Interestingly, the protein complex ZmRBR3/4/E2F has a unique role not observed in other plants or animals: high levels of ZmRBR3/4 in complex with E2F promotes the expression of genes involved in DNA replication and cell cycle progression; contrary to what has been reported for overexpression of pRb and AtRBR (Figure 2C) [118,119,123,124]. Besides, ZmRBR3/4 is negatively regulated by the ZmRBR1-E2F complex; as in lines expressing RepA, which inhibits ZmRBR1, ZmRBR3 is upregulated, suggesting a compensatory mechanism to ensure cell cycle progression [118,125]. This mechanism has also been observed in undifferentiated germline cells in which the absence of pRb is compensated by pRBL1 (p107) to maintain the cell’s quiescent state [126]. 

The conservation in plants and animals of the “restrictions point” and of key cell cycle regulators like Cyclin-CDKs and RB/E2F, suggests that the common ancestor of these two groups of organisms already had these components. Interestingly, there is a different number of genes encoding for each cell cycle protein suggesting the emergence of different and novel lineage-dependent proteins between plants and animals [78]. For example, it is known that Arabidopsis has 10 genes that code for cyclin-D (CYCD) classified into seven subtypes [127,128], while in humans there are only three [129]. In contrast, nine E2F are found in humans (three activators and six repressors), and only six in Arabidopsis. Of these, AtE2Fa functions as activator, AtE2Fb functions either as an activator or repressor, depending on the plant developmental stages; E2Fc is a repressor; and the other three members of this family (E2Fd/e/f) are atypical since they have a duplicated DNA-binding domain, do not heterodimerize with DP and lack the trans-activating and Rb pocket-binding domains, and thus resemble E2F7/8 in animals [130,131,132,133]. Finally, some components of the cell cycle regulatory machinery are plant-specific such as type B cyclin dependent kinases (CDKB1/2) [134,135], while others are animal-specific, such as cyclin E in humans [136,137].

The DREAM complex also appears to be conserved in plants [138,139]. In addition to the presence of AtRBR, E2Fs/DPs, a MuvB-like complex has also been found that contains ALY2/3 (orthologs of LIN9) and TCX5 that is part of the TSO1-like family members (orthologs of LIN54) [140,141]. Furthermore, MYB3R, a transcription factor of the Myb family, has a protein structure resembling the DNA binding domain of B-Myb, which in humans forms part of the MuvB complex. These plant’s MYB3Rs also participate in the regulation of the G2/M transition as follows: MYB3R3 associates with E2Fc and AtRBR to repress G2/M genes, while MYB3R4 associates with E2Fb and AtRBR to activate G2/M genes (Figure 2B) [142,143,144]. CDKA;1 and cyclins could as well be involved in the plant DREAM complex since MYB3R3 and MYB3R4 interact with CDKA;1 and, in tobacco, CDKA;1 regulates MYB3R phosphorylation [139,144,145]. Moreover, E2Fc can physically interact with CDKA;1, CYCD2;1, and CYCD2;2 in vitro (Figure 2B) [146]. Interestingly, the function of the activation complex MYB3R4/AtRBR1/E2Fb is similar to the function of ZmRBR3 in maize that positively regulates transcript accumulation of genes involved in DNA replication and cell cycle progression in the transition G2/M (Figure 2B,C) [124]. However, in contrast to what has been reported in animals, RBR presence in the DREAM complex is able to control cell cycle in different stages of cell cycle in plants (Figure 2B,C).

## 4. The Roles of Retinoblastoma in Cell Differentiation

The formation of any organ relies on two different but interlinked cellular processes: cell proliferation and cell differentiation. Proliferation produces all the cells that later will acquire fates, specialized functions, and morphologies through differential gene expression during cell differentiation [147,148]. pRb has been widely studied in proliferation and, recently, its participation in many different animal differentiation processes in the eye, brain, peripheral nervous system, muscle, liver, placenta, lung, cerebellum, pituitary gland, and heart has been elucidated (Figure 3A) [149,150,151,152,153,154]. 

The participation of pRb in these processes has been studied in vivo in mutant mice and/or cell cultures derived from cancer cells, characterized by alterations in their *RB1* expression levels [155,156,157]. Mice with *RB1* ablation are embryonic lethal, and those with low levels of the three pRb (*RB1*, *RBl1*, and *RBl2*) not only die in utero but also present defects in erythroid, neuronal, and muscular differentiation [158,159,160]. Additionally, in mice with conditionally-regulated levels of the three pRb, cells overproliferated inducing retinal cancer (Retinoblastoma) and also display defects in laminar organization of the retina and the crystalline [161,162,163]. E2Fs target genes are also important for differentiation of the adipose tissue, bone, nervous system, and muscles [164,165,166,167]. Interestingly, expression of pRb in mutants with altered E2F function has shown that pRb can also independently regulate tissue-specific genes in mammals, pointing to its broad roles in development [168,169]. 

Similar to pRb in animals, RBR participates in differentiation processes in the root and shoot meristems, the vascular system, leaves, stomas, and trichomes tissues in plants (Figure 3A) [116,117,170,171]. In Arabidopsis, *AtRBR* functions as a negative regulator of primary root development; its downregulation leads to longer roots with larger meristems whereas its overexpression results in shorter roots with smaller meristems [172]. In this organ, AtRBR forms a protein complex with the TF ARABIDOPSIS RESPONSE REGULATOR12 (ARR12) that activates the transcription of the *AUXIN RESPONSE FACTOR19* (*ARF19*) TF [172]. These TFs participate in two different hormone signaling pathways: ARR12 is part of the cytokinin signaling pathway involved in root differentiation, while ARF19 is part of the auxin signaling pathway which is important for root cell proliferation [172,173]. *ARF19* is activated by RBR mostly in between the meristematic and the elongation zones in roots and it is suggested to promote differentiation [172].

The role of RBR in differentiation at the shoot apical meristem (SAM) has also been analyzed. In this case, overexpression of AtRBR accelerates differentiation, increasing the expression of genes involved in metabolic pathways that are not present in the SAM and decreasing the expression of genes that are only expressed in the SAM [114,170]. *AtRBR* RNA interference (RNAi) in leaf primordia delays differentiation and, consequently, increments two to four-fold the cell number in both the adaxial and abaxial sides of the leaves [114,170]. In addition, also in leaves, AtRBR participates in the endoreplication and differentiation of trichomes, in which the AtRBR transcription is regulated by the TFs GLABRA1 (GL1) and GLABRA3 (GL3) [174,175]. In the vascular system, AtRBR associates with XYLEM NAC DOMAIN1 (XND1), a negative regulator of differentiation, thus controlling processes related to the differentiation of tracheary elements [176,177].

It is difficult to compare the roles of pRb and RBR during differentiation due to lack of information and because the components involved in cell differentiation are more species-specific and less conserved than those involved in cell cycle regulation. However, taking as examples skeletal muscle differentiation in humans and stomatal guard cells differentiation in Arabidopsis, conserved factors that coordinate cell differentiation can be identified, namely Rb, MADS-box TFs, and TFs of the bHLH family, as well as the cyclins/CDKs protein complexes (Figure 3D) [178,179].

At post-embryonic stages, myocyte differentiation in mammals can be triggered in response to muscle damage or a specific growth-inducing stimulus. This last provokes massive proliferation of myoblasts through the hierarchical activation of several MRF (Myogenic Regulatory Factors): Myogenic Factor 5 (Myf5), myoblast determination protein 1 (MyoD), Myogenin (MyoG), and Myogenic Regulatory Factor 4B (MRF4), which are TFs that belong to the bHLH family. These MRF are required together with pRb for the muscle regeneration process [180,181,182]. In resting state conditions, muscle stem cells or satellite cells express only the box protein Pax7 [183]. When the muscle is injured, satellite cells get activated and become myoblasts that massively proliferate to generate the myogenic progenitors, which express Myf5 and/or MyoD TFs [184]. Later, when myoblasts start differentiation into myocytes, Myf5 and Pax7 are repressed and MRF4 and myogenin are expressed promoting cell cycle exit. Finally, when the cells fuse to become myofibrils, myogenin and MRF4 are expressed as part of the final steps of differentiation (Figure 3B) [180,182]. In this process, the protein complex of MyoD with pRb, is thought to initiate differentiation, as it promotes cell cycle arrest [185,186,187,188]. Despite immunoprecipitation experiments that proved that pRb and MyoD can interact [189], later experiments using nuclear magnetic resonance allowed to determine that there is no direct protein–protein interaction between MyoD and pRb [190]. Still, MyoD does activate *RB1* expression through its association with the cAMP response element-binding (CREB) TF and the coactivators p300 and P/CAF (Figure 3) [191,192]. Myocyte Enhancer Factor 2 (MEF2), that belongs to the MADS-box family of TFs, is also a component necessary to carry out the final stages of muscle differentiation [149,193,194].

Induction of muscle biogenesis also requires the regulation by cyclins-CDKs in association with pRb [195]. The stable repression of cyclin D1, required for cell cycle arrest during differentiation, is regulated by the joint action of MyoD and pRb through the regulation of the upstream intermediary gene Fra-1 (antigen related to FOS 1) (Figure 3B) [196,197]. Antagonistically, cyclin D1 inhibits the activity of MyoD: overexpression of cyclin D1 promotes nuclear accumulation of CDK4, that binds MyoD, preventing its interaction with DNA, and inhibiting the CDK4-dependent phosphorylation of pRb (Figure 3B) [198,199,200]. Additionally, when HDAC1 interacts with pRb, the MyoD protein can bind its target DNA regulatory sites [201,202]. In summary, there are two different protein complexes formed during different stages of muscle development: HDAC1/MyoD during proliferation and HDAC1/hypophosphorylated pRb during differentiation (Figure 3B) [201,202]. 

In plants, some epidermal cells undergo differentiation producing the two mature guard cells that form the stomata pores, structures that are conserved among land plants and allow them to regulate gas exchange and water loss [203,204]. Stomatal development is hierarchically regulated by the sequential activation of several TFs of the bHLH family: SPCH (SPEECHLESS), MUTE, and FAMA. These three bHLHs form heterodimers with either the bHLHs SCREAM (SCRM, also called ICE1) or SCRM2 [205,206,207], that belong to the same family of TFs that participate in muscle development in mammals (MRFs) (Figure 3D). These plant TFs orchestrate cell division events of protodermal cells, which give origin to guard cells (stomatogenesis). SPCH triggers the maturation of a protodermal cell into a meristemoid mother cell (MMC) and is also involved in the asymmetric cell division of the MMC, that results in one meristemoid cell and one larger sister cell (SLGC). The meristemoid cell exits stemness and engages in differentiation to become a guard meristemoid cell (GMC). MUTE must be expressed at this stage, to direct further differentiation of a GMC, and to ensure that this cell undergoes a single symmetric division. In addition MUTE regulates the expression of *FAMA*, which controls the final stages of differentiation, promoting guard cell (GC) identity acquisition and the irreversible termination of the meristematic activity of the cells (Figure 3C) [208,209,210]. AtRBR plays important roles in the regulatory network of stomata development, and its downregulation at the GMC or GC stages, induces extra divisions in differentiated GCs and the formation of aberrant stomata-in-stomatal nested structures [114,211]. In fact, AtRBR hyperphosphorylation inhibits stomatal initiation affecting the asymmetric division of protodermal cells that produces MMCs, this seems to be controlled by CDKA;1, that negatively regulates AtRBR, and regulates positively SPCH TF through phosphorylation (Figure 3C) [212,213,214]. It has also been hypothesized that AtRBR hyperphosphorylation by CDKB1;1-CYCD7;1 inhibits the AtRBR/FAMA repression complex leading to the induction of cell-cycle regulators of the GMC symmetric division event [215,216,217]. At the same time, MUTE directly upregulates *FAMA* and *FLP;* and FAMA represses cell-cycle control genes such as CDKB1;1, ensuring a single symmetric division to form GCs (Figure 3C) [208,218]. Mutation in the FAMA LxCxE sequence prevents the formation of the AtRBR/FAMA complex, making cells unable to maintain the long-term commitment to differentiate into GC, and arresting this process at the GMC stage [211,215]. A similar mechanism is present in mammals, in which downregulation of MyoD allows cells to dedifferentiate; an ability that determines the muscular capacity to regenerate [219]. Finally, it is also possible that FAMA functions at early steps of guard cell differentiation since the AtRBR/FAMA heterodimer binds to *SPCH* and *FAMA* promoters, and this complex negatively regulates the accumulation of the *SPCH* transcript, which is normally expressed at early stages of guard cells development (Figure 3C) [211,220].

As it can be appreciated, the differentiation processes of skeletal muscle in mammals and guard cells in Arabidopsis are both regulated by a similar set of conserved elements: pRb/RBR, bHLHs, cyclins and CDKs (Figure 3D). Moreover, in both organisms, members of the MADS-box family participate in these processes: MEF2 is involved in early stages of muscle differentiation and in cell proliferation of other tissues (Figure 3B) [221,222,223]. Similarly, a plant MADS-box gene, *AGAMOUS-LIKE16* (*AGL16*), is expressed and participates in GC development (Figure 3D) [224,225]. AGL16 mediates the stomata development process at the MMC cell lineage level and represses FAMA as well as other genes involved in the development and differentiation of guard cells (Figure 3C) [225,226]. 

## 5. Retinoblastoma Function in Stem Cells Homeostasis

Accumulated evidence indicates that pRb loss of function in mammals results in altered progenitor cells (or stem cells) quiescence. Quiescent cells are usually in the G0 phase of the cell cycle or in a prolonged G1 phase; and have a very low proliferative activity. Through asymmetric division stem cells can give origin to a new quiescent stem cell and a new daughter cell that can proliferate several times to eventually produce one or more differentiated cell types [227,228,229]. pRb participates in pluripotency maintenance, inhibition of differentiation, and in self-renewal of stem cells [230,231,232]. Mutants with low expression levels of Rb display an increased cell division of both embryonic and post-embryonic stem cells from retina, mesenchymal, and early osteoblasts progenitors, as well as of post-embryonic stem cells of the liver, muscle, and nervous system [233,234,235,236,237,238,239]. Interestingly, in these pRb mutants somatic cells acquire stem cells features, as it occurs in some human cancers [240,241]. In human pluripotent stem cells (hPSC), the pRb/E2F pathway enhances differentiation towards all germ layers in response to a DMSO stimulus [153]. pRb together with E2F1 can bind and suppress the transcript accumulation of pluripotency promoting factors, such as *SEX DETERMINING REGION Y-BOX 2* (*SOX2*) and *OCTAMER-BINDING TRANSCRIPTION FACTOR 4* (*OCT4*). In addition, pRb can alter the accumulation of transcripts by directing chromatin modifiers to promoters of specific TFs, as it happens in the case of *KRUPPEL-LIKE FACTOR 4* (*KLF4*), the homeobox *NANOG* and *TRANSCRIPTION FACTOR 3* (*TCF3*), that are part of the induced stem cell pluripotency regulatory network; and also in the case of *ENHANCER OF ZESTE HOMOLOG 2* (*EZH2*), which is a methyltransferase that participates in establishing stem cells establishment [242,243].

The plant ortholog, RBR, is also implicated in pluripotency maintenance. Lowering transcript levels of *AtRBR* induces disorganization of the root stem cell niche (SCN) in Arabidopsis, as well as in the shoot meristem that harbors flower and leaf stem cells [114,115,116]. The root SCN consists of a central organizer, the quiescent center (QC), which is surrounded by four type of stem cells called initial cells, that in Arabidopsis are columella initials (CI), cortex/endodermis initials (CEI), epidermis/lateral root cap initials (ELRCI), and stele initials (SI) [244]. Under normal growth conditions, proliferation rate at the QC is very low compared to adjacent zones, although the division rate increases at the QC in older seedlings [245,246]. *AtRBR* silencing increases the proliferation of both QC and CI cells, resulting in an overgrowth of undifferentiated cell layers. Conversely, overexpression of *AtRBR* causes premature differentiation of CIs [115,116,247]. Interestingly, absence of *AtRBR* favors the duplication of differentiated columella cells that normally do not duplicate in wild-type plants [247].

Within CEI cells and their progeny, the TF SCARECROW (SCR), a member of the GRAS TFs family, interacts, through its LxCxE motif, with AtRBR forming a ternary complex with SHORTROOT (SHR), which is another member of the GRAS family. This AtRBR/SCR/SHR complex inhibits the transcription of target genes of the SHR/SCR heterodimer. One of these target genes is *CYCLIN D6;1* (*CYCD6;1*), which controls the cell cycle progression of the CEI cells progeny. Moreover, CYCD6;1 together with the kinase CDKB1;1, in turn, regulates the phosphorylation of AtRBR, liberating the SCR/SHR complex, favoring in this way CEI cells’ asymmetric division [115,248,249]. AtRBR, SCR, and CYCD6;1 are degraded by the proteasome before mitosis, which is consistent with a model where the degradation of these proteins allows CEI cells to restart the asymmetric divisions [248]. Moreover, the same AtRBR/SCR interaction is necessary to establish the QC cells and is regulated by the interaction of AtRBR with the ETHYLENE RESPONSE FACTOR115 (ERF115) TF, that belongs to the AP2/ERF family. This interaction also occurs through the LxCxE domain of ERF115, that competes with SCR for AtRBR, reducing in this way the levels of the AtRBR-SCR heterodimer [250]. Importantly, most of the AtRBR functions related to differentiation and cell cycle arrest in the SCN are cell-autonomous, highlighting the crucial role of AtRBR activity level in the QC, in the columella stem cells, and in their immediate progeny in the acquisition of niche-specific features [247].

Interestingly, it has also been observed that the TOPOISOMERASE 1α (TOP1α) together with AtRBR, is also involved in the control of stem cell maintenance during root development. TOP1α is an enzyme present in plants and animals that creates breaks in double DNA strands to relax supercoiled structures. In humans, it has been used as a target to stop the proliferation of breast cancer stem cells in cell cultures [251,252]. In Arabidopsis roots it has been shown that *TOP1α* is downregulated by AtRBR to maintain the undifferentiated state of cells and the number of CI cells in the SCN. In addition, *TOP1α* is epistatic over *AtRBR* and its overexpression results in an increased number of CI cells, as it happens in *AtRBR* loss of function mutants [253]. Besides, mutations that disrupt the activity of TOP1α, induce cell death in the initial cells of the stele (SI) which can be partially reversed by the activation of *ERF115* expression, since TOP1α negatively regulates the expression of *ERF115* [253], and as it was mentioned above, ERF115 also interacts with AtRBR to establish the QC [250]. Although it is not yet known whether TOP1α participates with AtRBR in the shoot meristem or not, TOP1α also regulates the establishment of stem cells through indirect transcriptional repression of *WUS* (*WUSCHEL*) [254].

In the shoot meristem, silencing of *AtRBR* also produces disordering of the stem cell divisions within the SAM, resulting in a significant reduction in the typical height-to-width ratio of the SAM; and also in alterations in stem cell maintenance and differentiation [114]. Overexpression of AtRBR triggers cells toward a more differentiated state; but the molecular mechanism has not yet been described to explain this phenotype [170]. In addition, AtRBR regulates proliferation and differentiation of MMC of guard cells in leaves as it has been described above.

It can be noted from the examples presented, that pRb/RBR of animals and plants, respectively, participate in the maintenance of stem cells by controlling the cell cycle and cell differentiation, as well as regulating specific genes that give identity to these cells.

## 6. Function of Retinoblastoma in Epigenetic Modifications, Chromatin Regulation, and DNA Damage 

Epigenetic modifier proteins participate in the regulation of gene expression throughout development. This allows cells with the same genetic background to exhibit different phenotypes [255,256]. Epigenetic modifications alter DNA accessibility and chromatin structure by mechanisms such as DNA methylation and histones modifications by acetylation, methylation, ubiquitination, phosphorylation, sumoylation, etc. [257,258]. In mammals and plants, many epigenetic modifier proteins interact with protein complexes that include pRb/RBR, allowing them to regulate different developmental processes.

Human pRb has been reported to interact with over 300 proteins and many such protein interactions are epigenetic modifier proteins or interact with the latter [40,256]. pRb levels decrease leads to an incomplete chromosome condensation and segregation during mitosis, as it has been observed in cancer cells; some alterations of the chromatin structure are also induced by changes in histone methylation and acetylation [259,260,261]. pRb can interact with chromatin remodeling factors, such as histone deacetylases, DNA methyltransferases, histone methyltransferases, and with complexes like SWI/SNF; and like Polycomb group (PcG), the latter is a chromatin-modifying complex that maintains repressed gene expression states and is subdivided into two main complexes: Polycomb repressive complex 1 (PRC1) and PRC2 [259,262,263,264,265].

The interaction of pRb-epigenetic with modifiers complexes are also important to maintain heterochromatin in intergenic zones as well as in centromeres and telomeres (Figure 4A) [260,261]. The interaction of pRb with ENHANCER OF ZESTE HOMOLOG 2 (EZH2), a histone methyltransferase of the PRC2 complex, allows the deposition of the trimethylation of lysine 27 of histone H3 (H3K27me3), a repressive mark, on pRb target genes (Figure 4A). In turn, pRb-E2F negatively regulates *EZH2* transcript accumulation and proliferation; conversely high expression of *EZH2* is observed in cancer stem cells as has a critical function in regulating stem cell expansion and maintenance (Figure 4A) [242,266]. Interestingly, it has been observed that pRb can also recruit EZH2 protein into sequences within introns and intergenic regions, specifically in repeated sequences, transposons, long interspersed nuclear elements (LINEs), short interspersed nuclear elements (SINEs), and long terminal retroviruses (LTR). The loss of the pRb-EZH2 complex provokes loss of the H3K27me3 mark at these elements, leading to dispersion or loss of heterochromatin and probably disorganized proliferation as observed in cancer cells (Figure 4A) [267]. 

DNA integrity in mammals is altered when pRb is absent and, in some cases, this can be provoked by the overexpression of E2F regulated genes that are able to introduce double-strand DNA breaks, or by stress conditions that generate aneuploidies [37]. In the case of a double strand break, pRb is necessary to form the complex of the heterodimer E2F1-pRb with TopBP1 (DNA topoisomerase 2-binding protein 1) [268]. TopBP1 is a protein that interacts with Topoisomerase 2β (Top2β) and with other proteins that participate in DNA replication and in the maintenance of the DNA integrity and genome stability [261,269]. In addition, the protein complex E2F1-pRb-TopBP1 interacts with BRG1 (also known as ATP-dependent chromatin remodeler SMARCA4) which is a member of the SWI/SNF complex that is necessary to reduce nucleosome density at injury sites, allowing DNA end resection and reparation by homologous recombination (HR) (Figure 4C) [268,270]. Another novel aspect of the pRb-BRG1 interaction is its influence in mediating cell cycle arrest, by the regulation of different genes also involved in human cancer cells (Figure 4C) [271,272,273]. pRb interacts with the tumor suppressor BRCA1 (Breast cancer 1), which is also involved in DNA repair via Homologous Recombination. The BRCA1-pRb complex interacts with histone deacetylases (HDAC1/2) and with topoisomerase 2β (Top2β) to regulate DNA stability (Figure 4C) [274,275]. Furthermore, the pRb-BRCA1 complex is involved not only in the response to DNA damage, but also in cell cycle control, as deletions in the BRCA1 binding domain with pRb, inhibits BRCA1-dependent cell cycle progression [262]. pRb also participates in another DNA repair pathway: the non-homologous end joining (NHEJ), the exact role of pRb in this mechanism is unknown but it has been reported that pRb interacts with two of the proteins that recognize the breakdown of the double chain: KU-70 and KU-80 (Figure 4C) [261,276].

In plants, there are also numerous reports of RBR interactions with epigenetic modifiers, which are important in the regulation of different developmental processes [277]. In Arabidopsis, like in animals, it has been shown that PRC2, a subcomplex of PcG, participates together with AtRBR in the establishment of the H3K27me3 mark during differentiation and development of the female and male gametophytes, in leaf development and during the establishment of stoma cell lineages. In these three processes, AtRBR associates with components of the PRC2 repressor complex such as MULTICOPYSUPPRESSOR OF IRA1 (MSI1), FERTILIZATION INDEPENDENT ENDOSPERM (FIE), VERNALIZATION 2 (VRN2), and CURLY LEAF (CLF), a gene orthologous to EZH2 from humans (Figure 4B) [66,278]. In addition, AtRBR together with MSI1 directly represses the expression of the DNA methylase *METHYLTRANSFERASE 1* (*MET1*) (Figure 4B), that maintains DNA methylation during DNA replication and regulates gene imprinting. The repression of *MET1* by this complex allows the transcriptional activation of *FERTILIZATION INDEPENDENT SEED 2* (*FIS2*) and *FLOWERING WAGENINGEN* (*FWA*), that are important for female gametogenesis [66,113,279]. In turn, *MET1* is positively expressed during male gametogenesis; and is important for maintaining the gene repression of *FIS2* and *FWA* in the paternal allele, leading the monoparental expression of these genes during fertilization and endosperm development, [279,280]. Furthermore, *AtRBR* loss of function mutants present higher levels of *SWINGER* (*SWN*), *MSI1*, and *FIE* transcripts, which are components of the PRC2 complex. Interestingly, the *AtRBR* transcript in pollen is directly repressed by the PRC2 complex (Figure 4B) [113]. In plant embryos, the PRC2 complex with AtRBR directly binds and deposits the H2K27m3 mark on different embryonic genes, leading to their repression and subsequent seed germination (Figure 4B) [69,281]. Similarly, in stomatal development, AtRBR/FAMA heterodimer is required to recruit PRC2 to H3K27me3 deposition into *SPCH* and *MUTE* regulatory regions, and repress its transcript accumulation, necessary to control differentiation and stomatal development correctly (Figure 4B) [211,215,282].

AtRBR also appears to regulate DNA repair in several conditions. First, AtRBR binds and represses genes involved in homologous recombination such as *RADIATION SENSITIVE 51* (*RAD51*) and helps to locate RAD51 to the right place at DNA lesions (Figure 4D) [283]. Additionally, TOP1α is critical to ensure genome integrity and survival of root stele stem cells, as the loss of function of *TOP1α* triggers DNA double-strand breaks and cell death in these cells; in the root, *TOP1α* is downregulated by AtRBR (Figure 4D) [253]. Although the participation of AtRBR and Top1α in the shoot meristem have not yet been studied, TOP1α participates with the PRC2 complex in the repression of the *WUS* locus (Figure 4D) [252,253,254]. 

AtRBR also is recruited to damaged DNA sites, along with E2Fa and AtBRCA1 and helps to maintain the integrity of the root meristem (Figure 4D). Furthermore, similar to what is observed for animals for BRCA1 and pRb (Figure 4C) [262,284], AtRBR and AtBRCA1 have been shown to physically interact when cells are damaged [285]. In addition, E2Fa is required for *AtBRCA1* expression, when genotoxic stress is induced (Figure 4D) [285]. Thus, it would be interesting to analyze if the AtBRCA1-AtRBR complex participates in the regulation of the cell cycle, as it occurs in humans. Finally, analysis of chromosome sites to which AtRBR physically binds, show that this protein not only targets gene regulatory sequences, but also transposons, especially Miniature Inverted-repeat Transposable Elements (MITEs) (Figure 4B) [142].

## 7. Conclusions

Development is a process where proliferation and differentiation cellular rates must be finely regulated. As we can appreciate from the examples presented throughout the text, pRb/RBR are multifunctional and highly connected proteins that control cell fate determination and differentiation through interactions with different proteins. The pRb/RBR structures and diverse post-translational modifications allow the proteins to differentially interact with an exceptionally high number of proteins, making them a key node in several regulatory networks. Interestingly, many of the protein partners are conserved between animals and plants, and in both lineages are involved in equivalent cellular processes such as cell cycle regulation, stem cell homeostasis, and cell differentiation. In addition, interaction with epigenetic and DNA topology regulators suggests that the protein–protein networks that involve RETINOBLASTOMA are also similar in plants and animals. Thus, important aspects of the regulatory networks underlying cell proliferation and differentiation in which this protein is involved, seem to be shared by plants and animals, despite the fact that these two lineages have unique cellular and structural characteristics.

## Figures and Tables

**Figure 1 ijms-21-04925-f001:**
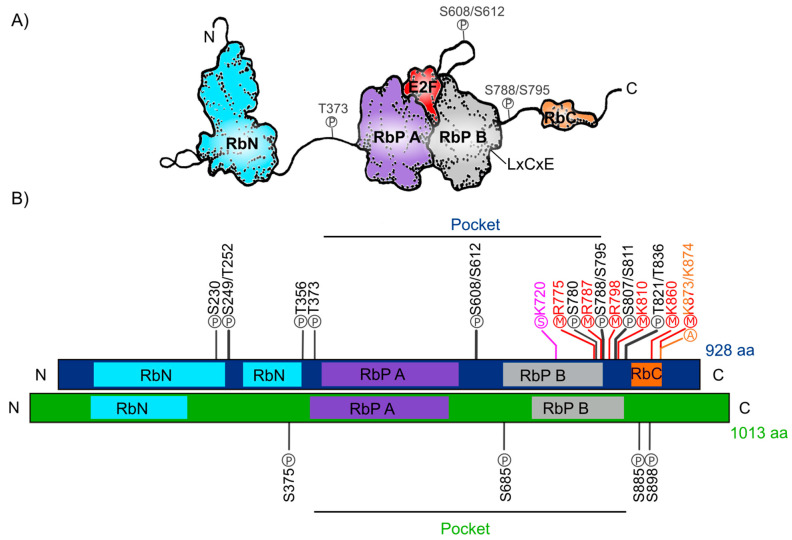
Retinoblastoma protein structure. (**A**) Representation of the human Rb protein structure with the domains RbN (blue), Pocket (RbP), with the RbP A (purple) and B (grey) subdomains interacting with an E2F TF (red), the RbC domain (orange) and the inter-domains (black lines) are also shown. The “P” inside a circle represents three examples of phosphorylation sites that change the structure of the protein. The position of the LxCxE cleft that allows Retinoblastoma protein (pRb) to interact with different proteins is also shown. (**B**) Comparison of the domains of human Rb protein (blue foreground) and Arabidopsis RBR protein (green foreground) and their reported post-translational modifications. Phosphorylation sites (P) are shown in black, methylation sites (M) in red, acetylation sites (A) in orange, and sumoylation sites (S) in pink.

**Figure 2 ijms-21-04925-f002:**
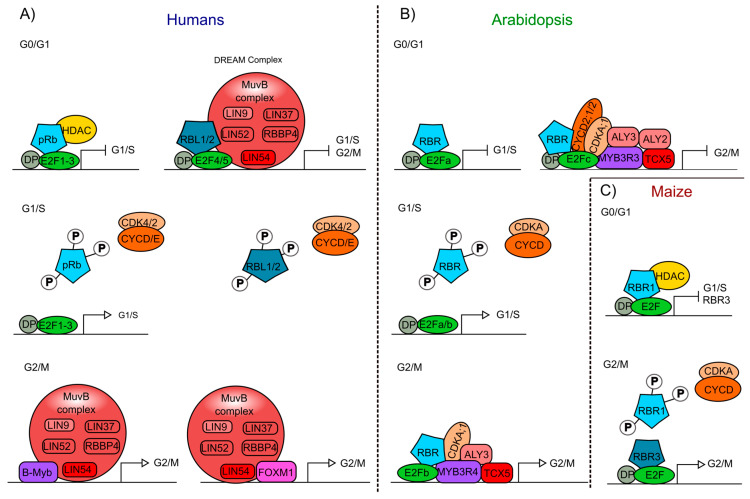
Interaction of the Retinoblastoma protein involved in cell cycle regulation of humans and plants. (**A**) pRb- and pRBL1/2-containing protein complexes from humans, formed at different stages of the cell cycle (G0/G1; G1/S; G2/M). (**B**) Arabidopsis protein complexes formed at different stages of the cell cycle (G0/G1; G1/S; G2/M), including AtRBR as a component. (**C**) Maize protein complexes formed at different stages of the cell cycle (G0/G1; G2/M), involving the ZmRBR proteins (RBR1 and RBR3) as components. Similar components in humans, Arabidopsis, and maize are displayed using the same colors: Rb proteins (blue), E2F transcription factors (TFs) (green), DP (grey), cyclins (CYC), and cyclin-dependent kinases (CDK) (orange), Muv complex proteins (red and pink), Myb TFs (purple), FOXM1 TF (pink).

**Figure 3 ijms-21-04925-f003:**
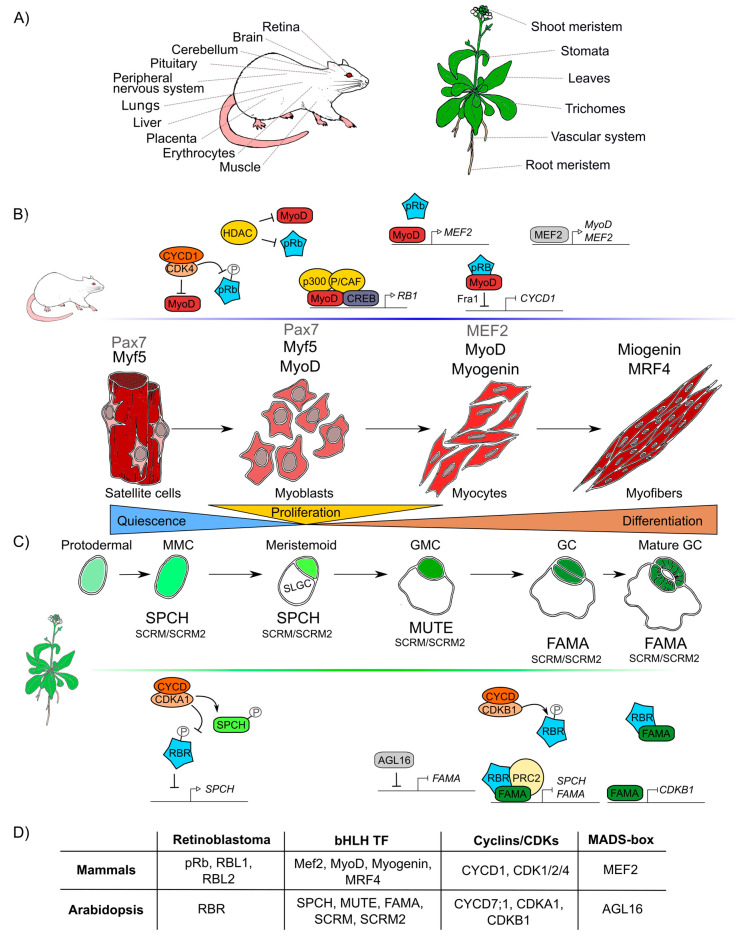
Retinoblastoma proteins are involved in cell differentiation in mammals and plants. (**A**) Mammals’ and plants’ organs whose differentiation depends on pRb/RBR. (**B**) Sequential steps of skeletal muscle differentiation in mammals. Shown are pRb interactions and bHLHs-family proteins (Mef2, MyoD, Myogenin, MRF4), involved in muscle quiescence maintenance, proliferation, and differentiation. (**C**) Sequential steps of guard cells differentiation in Arabidopsis. Shown are RBR interactions and bHLHs-family proteins (SPCH, MUTE, FAMA, SCRM, SCRM2) involved in quiescence maintenance, proliferation, and differentiation of guard cells. (**D**) Correlations between components involved in muscle and guard cell development in mammals and Arabidopsis, respectively. Differentiation in both lineages involves proteins of the pRb, bHLH TF, cyclins and CDKs, MADS-box TF families.

**Figure 4 ijms-21-04925-f004:**
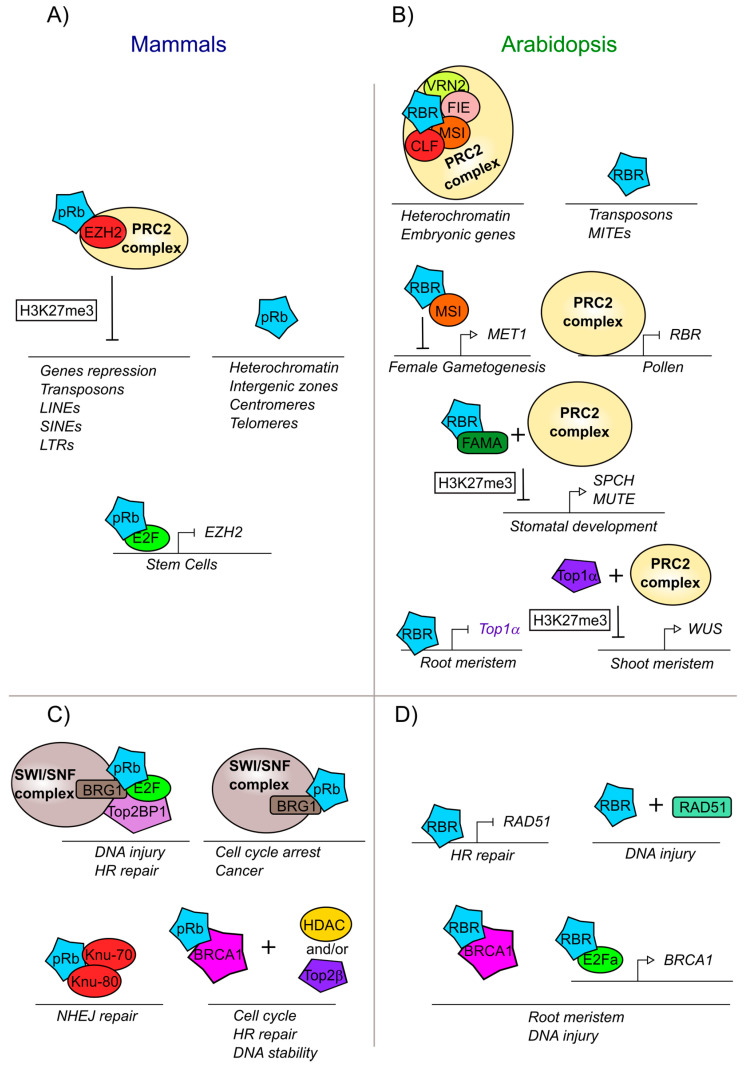
pRb and RBR are involved in modifications and DNA repair mechanisms both in mammals and plants. (**A**) Mammalian pRb participates in developmental processes and chromatin localization together with EZH2, a component part of the PRC2 complex. (**B**) Arabidopsis RBR participates in development and chromatin localization together with the PRC2 complex. (**C**) Mammalian pRb is part of the machinery involved in DNA repair. (**D**) Arabidopsis RBR is part of the machinery involved in DNA repair. Proteins and complexes conserved between mammals and Arabidopsis are marked with the same colors.

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
