# Peer review of "Beyond What Your Retina Can See: Similarities of Retinoblastoma Function between Plants and Animals, from Developmental Processes to Epigenetic Regulation"

_ijms, 2020, doi:10.3390/ijms21144925_

Round 1

Reviewer 1 Report

My comments for the review manuscript titled "Beyond what your retina can see: what plants can show about RETINOBLASTOMA function compared to what is known in animals" -

Understanding and impression:

Above titled manuscript reviews the literature discussing and highlighting parallels between retinoblastoma proteins from humans and theirs orthologs mainly from plants.  Authors describe these parallels in terms of pRb's protein structures and their  role in cell cycle control, cell differentiation, stem cell homeostasis, epigentic, chromatin regulation and DNA damage. 

Overall the literature review seems to be substantial for the topic at hand and write up is attempted to encapsulate the relevant information supporting proposed argument.  

However I have additional comments/suggestions and questions before the work can be considered for publication,

1) I think abstract write up is more general and can be further improve, or additional paragraph, with more specific info from paper so as to reflect scientific content of the review. 

2) There are some redundancies in introduction section. 

3) Readers would appreciate inclusion of paragraph describing clinical relevance of pRb from therapeutic standpoint. (CDk4/6 inhibitors etc) 

4) From the write up its not clear why there is need for studying plant orthologues of pRb protein ? any special advantages e.g. ease of studying plant systems vs humans ? and where is the bottleneck in our understanding of pRb proteins. 

5) What are the caveats of translating plant molecular biology findings and correlating with animal systems especially humans? 

Author Response

Understanding and impression:

Above titled manuscript reviews the literature discussing and highlighting parallels between retinoblastoma proteins from humans and theirs orthologs mainly from plants.  Authors describe these parallels in terms of pRb's protein structures and their  role in cell cycle control, cell differentiation, stem cell homeostasis, epigentic, chromatin regulation and DNA damage. 

Overall the literature review seems to be substantial for the topic at hand and write up is attempted to encapsulate the relevant information supporting proposed argument.  

Thank you very much for the overall positive evaluation of our manuscript. We enjoyed writing the paper and hope it will be of interest to the reader. 

However I have additional comments/suggestions and questions before the work can be considered for publication,

1) I think abstract write up is more general and can be further improve, or additional paragraph, with more specific info from paper so as to reflect scientific content of the review. 

Thank you very much for this observation, we have edited the abstract to make it more according to the content of the manuscript.

The Abstract now reads as follows:

The Retinoblastoma protein (pRb) is a key cell cycle regulator conserved in a wide variety of organisms. Experimental analysis of pRb’s functions in animals and plants has revealed that this protein participates in cell proliferation and differentiation processes. In addition, pRb in animals and its orthologs in plants (RBR), are part of highly conserved protein complexes which suggest the possibility that analogies exist not only between functions carried out by pRb orthologs themselves, but also in the structure and roles of the protein networks where these proteins are involved. Here, we present examples of pRb/RBR participation in cell cycle control, cell differentiation and in the regulation of epigenetic changes and chromatin remodeling machinery, highlighting the similarities that exist between the composition of such networks in plants and animals.

2) There are some redundancies in introduction section. 

Thank you very much for pointing this to us. We have cleaned the introduction to remove redundancies.

3) Readers would appreciate inclusion of paragraph describing clinical relevance of pRb from therapeutic standpoint (CDk4/6 inhibitors etc)

Thank you for this observation, we have added a paragraph in the introduction (page 2) section highlighting the clinical relevance of pRb: 

IntroductionMoreover, altered regulation of the pRb pathway is considered one of the most common traits in different types of cancer (Knudsen et al., 2019; Knudsen & Wang, 2010), and several studies have proposed targeting pRb regulation pathway as alternative treatments (Henry et al., 2019; Sachdeva & O’Brien, 2012; Tarang et al., 2020). In fact, Cyclin Dependent Kinases (CDKs), the kinases that phosphorylate pRb are commonly deregulated in many malignant tumors. From the therapeutic standpoint, pRb cannot be a target of drugs, however, CDKs are therapeutic targets, and several generations of non-specific cell cycle CDKs inhibitors have been under clinical evaluation as cancer treatments with mixed results. More recently specific cell cycle CDK4/6 and transcriptional CDKs inhibitors may become alternative therapeutic strategies under current clinical evaluation (Ding et al., 2020; Kwapisz, 2017; Li et al., 2019). In summary, a more thorough understanding of pRb’s developmental functions could help find new efficient treatments for different cancer types.

4) From the write up its not clear why there is need for studying plant orthologues of pRb protein? any special advantages e.g. ease of studying plant systems vs humans ? and where is the bottleneck in our understanding of pRb proteins. 

Thank you very much for this observation. We agree this is an important point, although it is probably wider than the case of pRb and its interactions. To determine those cases when an analysis of RBR in plants would be of use to infer processes related to pRb in animals, we need to know not only about similarities between pRb and RBR, but also about their respective interactions with other proteins. In this review, we opted to highlight such similarities between pRb and RBR interaction networks in animals and plants, and the similarity between the proteins composing them. This, in addition to the highly conserved character of pRb/RBR, and the processes they are involved in (proliferation, quiescence, differentiation), gives us a clue that RBR in plants can be used as a suitable model for the analysis of pRb interactions, and those diseases the latter is involved in. Of course, differences between animals and plants have to be expected, especially because of their fundamentally different strategies of differentiation and morphogenesis. We think that the extent to which plants can be used to analyze the function of pRb in animals will have to be analyzed experimentally. However, we already presented some examples when heterologous expression of animal proteins in plants results in functional complementation.

The principal bottleneck in studies of pRb (in animals) is related to the difficulties in predicting where proliferation/differentiation events will occur in the body. In contrast, in plants such zones have very specific locations within the body — they are located within meristems, where cell proliferation mostly takes place. Adjacent to this zone is the differentiation zone. Such organization makes from plants a model object that is very amenable to study these processes.

5) What are the caveats of translating plant molecular biology findings and correlating with animal systems especially humans? 

This question is partially related to the previous one. At the end of the day, plants and humans belong to different kingdoms, and thus, the applicability of findings made in plants will have to be tested in animals. We expect that higher functional similarity between pRb/RBR functions and interactions can be observed in processes that are common both for plants and animals. These processes are, for example, proliferation and quiescence. However, bigger differences may be expected in processes that differ in both lineages, for example — differentiation. But even for this last case we presented evidence that proteins belonging to the same families may be involved in this process (the cases of muscle and guard cell differentiation).

Also, as we have mentioned, plants share with humans more oncogenesis-related genes than Drosophila melanogaster and Saccharomyces cerevisiae. In addition, plants are multicellular organisms, and we illustrated the importance of pRb/RBR in appearance of this feature, which is required for the development. 

Finally, plants and particularly — Arabidopsis, have a prolonged postembryonic development with well localized sites of cell proliferation and differentiation — the meristems. Unlike animals (in which proliferation/differentiation sites are spread along the body), such organization in plants make it very easy to monitor processes of specification and morphogenesis, as well as determining the role RBR plays in them. We believe that despite the topological differences in the morphogenesis between plants and animals, the high similarity of the regulatory networks that control essential cell processes in plants and animals will make it possible to translate findings made in plants to the animal context.

Reviewer 2 Report

The authors present a REVIEW ARTICLE drawing comparisons between the mammalian and plant retinoblastoma protein. The review does a fairly through job at incorporating the many facets of pRB functions in the cell and comparing the two kingdoms. General comments: The manuscript should be revised by a fluent English speaker to polish spelling and grammatical mistakes, but in general it is quite comprehensible. There is an excess of references used, several of which are irrelevant to the statements that they are supporting. Suggest to revise these. In page 9, 3rd paragraph is a "(REF)" that was missed. The title and the abstract suggests that the manuscript will center in comparative analysis of the functions of pRB in animals and plants. However, the last paragraph in the introduction states "Therefore, in this review, we will focus on how RB1 or its plant ortholog gene RETINOBLASTOMA-RELATED (RBR) participates in important developmental processes such as: cell cycle control, cell differentiation, and in the maintenance of stem cells/pluripotency common to plants and mammals." There are areas in the manuscript were the description of the processes are written in great detail, in line with this statement, while other areas of the manuscript, the processes are glanced over and focused on the kingdom comparisons. The style should be consistent across the manuscript, and since the review is probably of most interest to people who are already familiar with pRB, perhaps the second approach (which is insinuated by the title) should be the one followed throughout. Introduce acronyms once, when it is first used in the manuscript and use the acronym thereafter. Content comments: page 2, second paragraph: "As determined by karyotyping, this disease is associated with recurrent losses on chromosome 13, which contains the RB1 gene." Chromosome 13 loss has been described in retinoblastoma tumors, but they are NOT recurrent. page 2, third paragraph: all that information is repeated from the previous paragraph. Delete. page 4, third paragraph: "For example, the S375 phosphorylation site in AtRBR is conserved in the same interdomain to the one found in pRb (S608/S612), suggesting a conserved function mediating the interaction with E2F TFs in animals and plants (Figure 1B) (69,71,72,74–76)." None of these references state that S375=S608/S612. In Figure 1B, these sites are not in the same interdomain. Please revise. page 4 last paragraph through the end of the section, the authors present the evolution of pRB. Evolution is not necessarily relevant for the topic and it distracts rather than aids in the topic presented by the title and abstract, recommended eliminating it. page 6, 1st paragraph: "The eukaryotic cell cycle process consists of four phases: Synthesis (S) and Mitosis (M), preceded by two interfaces (G1 and G2)..." The cell cycle has 2 major phases: Interphase and Mitotic phase. During Interphase there are 3 stages: G1, S, and G2. page 6, 2nd paragraph: describes process in yeast. The manuscript is on comparison between animal and plants, yeast introduced here and not mentioned anywhere else is unnecessary. Delete paragraph.  page 8, last paragraph: introduce the DREAM complex first. page 9, 2nd paragraph: RB1 mutation alone is sufficient to cause embryonic lethality and differentiation defects. page 15: "Another novel aspect of pRb-BRG1 interaction is its influence in the positive expression of different genes involved in human cancer cells (Figure 4C) (333–336)." This sounds like pRB-BRG1 drive the expression of genes when their action is transcriptional repression. It is the loss of pRB that breaks the pRB-BRG1 function leading to upregulation of genes that drive tumorigenesis.  REF#12, 71 are incomplete. Revise references.

Author Response

Comments and Suggestions for Authors

The authors present a REVIEW ARTICLE drawing comparisons between the mammalian and plant retinoblastoma protein. The review does a fairly through job at incorporating the many facets of pRB functions in the cell and comparing the two kingdoms. General comments: The manuscript should be revised by a fluent English speaker to polish spelling and grammatical mistakes, but in general it is quite comprehensible. 

Thank you very much for revising our manuscript. We have worked over it following your suggestions, and think we have improved it.

There is an excess of references used, several of which are irrelevant to the statements that they are supporting. Suggest to revise these

Thank you for this observation. We revised all references and selected the most appropriate ones.

In page 9, 3rd paragraph is a "(REF)" that was missed

Thank you for pointing us this error, we have corrected it. 

The title and the abstract suggests that the manuscript will center in comparative analysis of the functions of pRB in animals and plants. However, the last paragraph in the introduction states "Therefore, in this review, we will focus on how RB1 or its plant ortholog gene RETINOBLASTOMA-RELATED (RBR) participates in important developmental processes such as: cell cycle control, cell differentiation, and in the maintenance of stem cells/pluripotency common to plants and mammals." 

Thank you for this observation. We have changed the title and rewritten part of the abstract, introduction and conclusions to reflect the focus of our paper on the identification of similarities between pRb and RBR, and of similarities in the composition of the interaction networks of both these proteins. We wanted to highlight that similar regulatory networks participate in the regulation of cellular processes that are common for plants and animals. 

There are areas in the manuscript were the description of the processes are written in great detail, in line with this statement, while other areas of the manuscript, the processes are glanced over and focused on the kingdom comparisons.The style should be consistent across the manuscript, and since the review is probably of most interest to people who are already familiar with pRB, perhaps the second approach (which is insinuated by the title) should be the one followed throughout.

Thank you very much for this observation. As we mentioned above, we have opted to focus on the similarities between pRb and RBR and their interaction partners within regulatory networks. Particularly, we were interested in proposing plants as a suitable model to study pRb, and for that, we made a general comparison of the partners with which pRb and RBR interact. However, we are aware that, as plants and animals present differences, a specific case by case functional analysis will be required to determine whether findings made in plants can be translated to animals.

There is only one section (differentiation) in which we can perform such a comparison between plants and animals over time, as each of the analyzed structures develops. To the very best of our knowledge, this is the only example in which in-depth comparisons are possible, because of the availability of related information. In other cases the information is scarcer.

Introduce acronyms once, when it is first used in the manuscript and use the acronym thereafter. 

Thank you for pointing us this issue, we have corrected it.

Content comments: page 2, second paragraph: "As determined by karyotyping, this disease is associated with recurrent losses on chromosome 13, which contains the RB1 gene." Chromosome 13 loss has been described in retinoblastoma tumors, but they are NOT recurrent. 

Thank you for pointing us this error, we have corrected it.

page 2, third paragraph: all that information is repeated from the previous paragraph. Delete. 

Thank you for pointing us this issue, we deleted the repeated information.

page 4, third paragraph: "For example, the S375 phosphorylation site in AtRBR is conserved in the same interdomain to the one found in pRb (S608/S612), suggesting a conserved function mediating the interaction with E2F TFs in animals and plants (Figure 1B) (69,71,72,74–76)." None of these references state that S375=S608/S612. In Figure 1B, these sites are not in the same interdomain. Please revise. 

Thank you for this observation; the aminoacid site was wrong and we have corrected it. The aminoacids that we found that are in the same interdomain, between RbP A and RbP B subdomains in plants and animals are: S685 for RBR and S608/S612 for pRb. 

We also accept our error in the references: the ones you are quoting (Page 5 line 162 in the new version) belong to the previous sentence (Page 5 line 160 also in the new version) that talk about what is known about RBR protein and phosphorylation in plants. 

We have revised and corrected the references.

page 4 last paragraph through the end of the section, the authors present the evolution of pRB. Evolution is not necessarily relevant for the topic and it distracts rather than aids in the topic presented by the title and abstract, recommended eliminating it.

Thank you for this observation. We agree with you and have deleted the paragraphs concerning the examples of evolution from this section. We originally added this information because there are few Retinoblastoma reviews that include evolutionary studies. Especially, we wanted to emphasize that, despite the fact that both pRb and RBR evolved independently, the resulting structure and functions are very similar in plants and animals.

page 6, 1st paragraph: "The eukaryotic cell cycle process consists of four phases: Synthesis (S) and Mitosis (M), preceded by two interfaces (G1 and G2)..." The cell cycle has 2 major phases: Interphase and Mitotic phase. During Interphase there are 3 stages: G1, S, and G2. 

Thank you very much for pointing us this issue, we have corrected it. 

page 6, 2nd paragraph: describes process in yeast. The manuscript is on comparison between animal and plants, yeast introduced here and not mentioned anywhere else is unnecessary. Delete paragraph.  

We agree with this observation. We deleted the paragraph related to yeast.

page 8, last paragraph: introduce the DREAM complex first. 

Thank you very much for this observation. The Dream complex was introduced on page 6, last paragraph.

page 9, 2nd paragraph: RB1 mutation alone is sufficient to cause embryonic lethality and differentiation defects.

Thank you for this observation. We have added this new information: Mice with RB1 ablation are embryonic lethal, and those with low levels of the three pRb (RB1, RBl1 and RBl2) not only die in utero but also present defects in erythroid, neuronal and muscular differentiation  (Clarke et al., 1992; Jacks et al., 1992; Lee et al., 1992).

page 15: "Another novel aspect of pRb-BRG1 interaction is its influence in the positive expression of different genes involved in human cancer cells (Figure 4C) (333–336)." This sounds like pRB-BRG1 drive the expression of genes when their action is transcriptional repression. It is the loss of pRB that breaks the pRB-BRG1 function leading to upregulation of genes that drive tumorigenesis.  

Thank you for this information, we have corrected this in the manuscript and in the figure. “Another novel aspect of the pRb-BRG1 interaction is its influence in mediating cell cycle arrest, by the regulation of different genes also involved in human cancer cells (Figure 4C) (Dunaief et al., 1994; Marquez-Vilendrer et al., 2016; Strobeck et al., 2000).

REF#12, 71 are incomplete. Revise references. 

Thank you for pointing us this error, we have corrected it.

Reviewer 3 Report

This review by Zluhan-Martínez is a very comprehensive description of the literature and research on pRb in animals and RBR in plants. Cross kingdom analyses are certainly challenging and the authors cover 249 references to illustrate the parallels between plants and animals. There are some very minor grammatical errors that could be corrected but overall the manuscript is well written and clear with quality figures.

While I appreciate that comparative genetics and genomics can be a powerful way to leverage understanding from different species, I am not sure that I am persuaded by the argument that RBR research in Arabidopsis (let alone plants) will be useful for studying human disease and cancer in this case. (I found the differences between plants and animals interesting.) This argument would be more convincing if the authors could note specific examples of Arabidopsis research (perhaps as they arise in the various sections of the review) that could be applied to human cells. Or are there examples where this has already been achieved? Is the idea simply that if new factors (genes) or interactions are discovered in Arabidopsis, then homologues or analogous interactions could be investigated in humans? Or would this involve a more targeted approach? That said, the authors are entitled to present this view; especially as this is a review; this comment is just to note that more explicit detail would make it more persuasive.

Author Response

This review by Zluhan-Martínez is a very comprehensive description of the literature and research on pRb in animals and RBR in plants. Cross kingdom analyses are certainly challenging and the authors cover 249 references to illustrate the parallels between plants and animals. There are some very minor grammatical errors that could be corrected but overall the manuscript is well written and clear with quality figures.

While I appreciate that comparative genetics and genomics can be a powerful way to leverage understanding from different species, I am not sure that I am persuaded by the argument that RBR research in Arabidopsis (let alone plants) will be useful for studying human disease and cancer in this case. (I found the differences between plants and animals interesting.) This argument would be more convincing if the authors could note specific examples of Arabidopsis research (perhaps as they arise in the various sections of the review) that could be applied to human cells. Or are there examples where this has already been achieved? Is the idea simply that if new factors (genes) or interactions are discovered in Arabidopsis, then homologues or analogous interactions could be investigated in humans? Or would this involve a more targeted approach? That said, the authors are entitled to present this view; especially as this is a review; this comment is just to note that more explicit detail would make it more persuasive.

Thank you very much for this observation. We changed some points in the review to emphasize the similarities, and not the functional analogies, between the Rb function in plants and animals. We made more explicit the abstract, introduction and conclusions to be more persuasive that the similarities presented suggest that once regulatory networks were assembled during evolution, the key role of RETINOBLASTOMA as an integrator of internal developmental cues remained functionally constrained among eukaryotic organisms’ evolution; this also suggests the possibility that analogies exist not only between functions carried out by pRb orthologs themselves but also in the structure and roles of the protein networks where these proteins are involved. By highlighting the striking similarity between the architectures of pRb/RBR’s interaction networks, we propose that plants might be of use to obtain insights on pRb/RBR functions and interactions, although this knowledge should then be tested in animals to prove its viability.

We also added in the introduction some examples when Arabidopsis has served as a model organism to study drugs effect in humans: 

“Nonetheless, Arabidopsis thaliana (hereafter Arabidopsis), the most studied model plant, has been shown to be an important model system to understand basic regulatory mechanisms involved in human diseases (Jones et al., 2008; Spampinato & Gomez-Casati, 2012; Xu & Møller, 2011). For example, Arabidopsis has homologous genes for 70% of those involved in human cancer. Interestingly, a higher percentage than that found in the genome of Drosophila melanogaster or of Saccharomyces cerevisiae (Jones et al., 2008; Xu & Møller, 2011). Hence, Arabidopsis has already been used as a screening tool to evaluate the action and efficacy of some drugs to treat human cancer and other diseases (Papadia et al., 2017; Spampinato & Gomez-Casati, 2012; Vergara et al., 2015).”